# EVA-RNA: A Scaling Cross-Species Transcriptomic Foundation Models for Immunology & Inflammation

**Aziz Fouché, Yannis Cattan, Charlotte Claye, Matthew Corney, Pierre Marschall, Karim El Kanbi, Julien Duquesne**

Scienta Lab, Paris

## ABSTRACT

Recent studies have revealed that transcriptomic foundation models often fail to outperform simple baselines on clinically relevant tasks, suggesting a disconnect between pretraining objectives and useful representations. To bridge this gap, we introduce EVA-RNA, a transformer model pretrained on a curated corpus of over 500k samples spanning human and mouse, including bulk RNA-seq, microarray, and pseudobulked single-cell data, with a focus on Immunology & Inflammation. EVA-RNA exhibits clear power-law scaling across 7M to 300M parameters, with no sign of plateauing, in contrast to prior reports of diminishing returns in single-species models. Also, pretraining improvements consistently translate to downstream performance, as measured by a holistic benchmark spanning drug discovery, preclinical translation, and clinical applications. We finally conduct explainability experiments to explore (i) the concepts in EVA-RNA's representations, (ii) the structure of orthologous genes in latent space, and (iii) the evolution of intrinsic dimensionality across layers and throughout training.

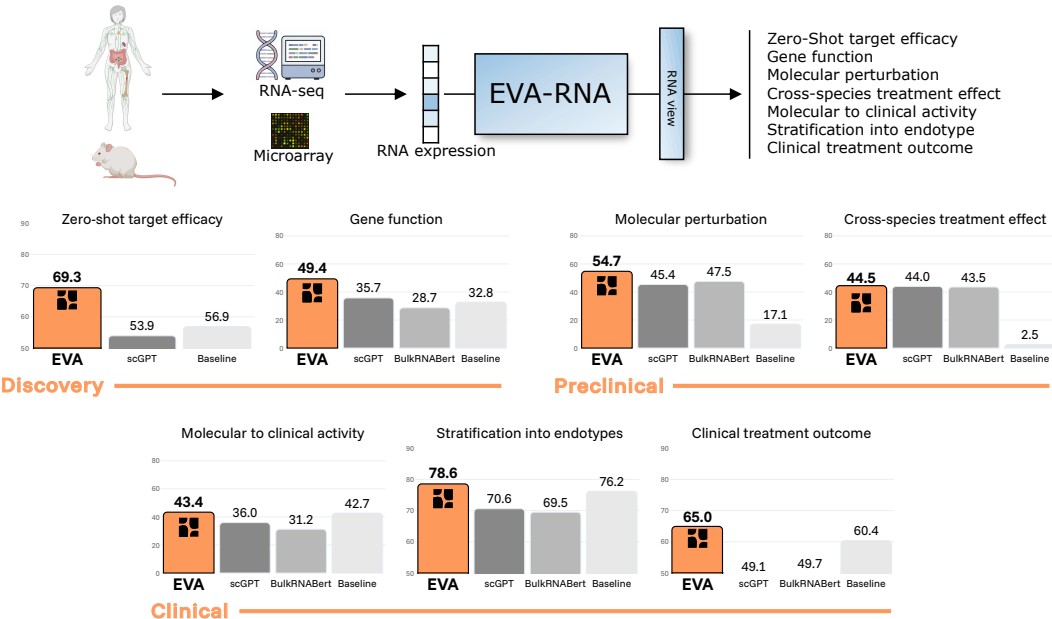

# 1 INTRODUCTION

Biological foundation models have emerged as a promising paradigm for learning rich representations from gene expression data (Theodoris, 2024). However, recent benchmarks reveal that these

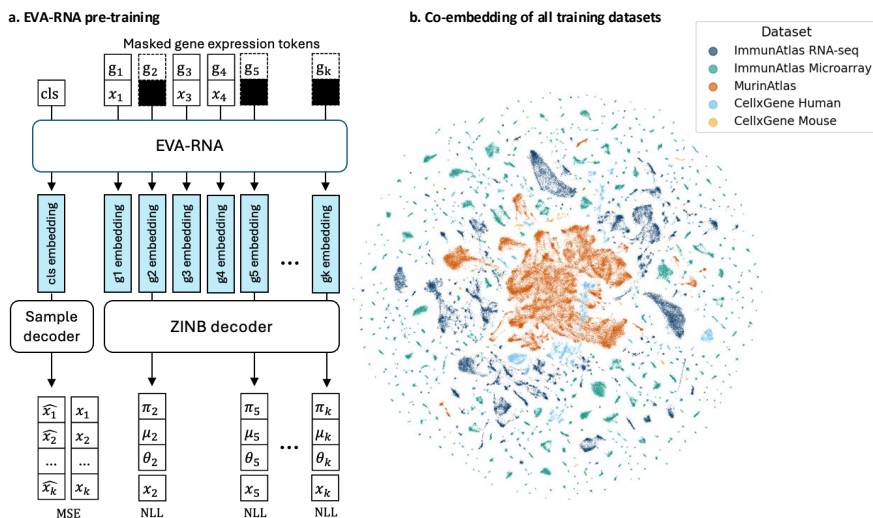

Figure 1: **EVA-RNA pretraining.** (a) EVA-RNA pretraining algorithm, combining stochastic masked expression prediction and full expression reconstruction from the CLS token embedding. (b) UMAP of the pretraining samples embedded through EVA-RNA's CLS token (300M).

models often fail to outperform simpler baselines on relevant downstream tasks, particularly in out-of-distribution scenarios (Kedzierska et al., 2025; Ahlmann-Eltze et al., 2025), exposing a possible misalignment between pretraining objectives and the representations required for effective transfer learning. Whether transcriptomic foundation models exhibit predictable scaling behavior analogous to large language models also remains an open question; AIDO.Cell (Ho et al., 2024), a multi-species single-cell foundation model, notably reported diminishing improvements in validation loss beyond 100M parameters.

A critical gap in many transcriptomic foundation models is their limited ability to integrate across species and technologies. Drug development relies heavily on translating findings from mouse models to human disease, yet existing models such as scGPT (Cui et al., 2024), Geneformer (Theodoris et al., 2023), and BulkRNABert (Gélard et al., 2025) were trained exclusively on human samples and often on a single transcriptomics modality. This restricts their applicability in translational research, where cross-species molecular translation is essential. Meanwhile, decades of microarray data remain underutilized due to the difficulty of integrating legacy platforms with modern RNA-seq.

We introduce EVA-RNA, a 300M-parameter transformer encoder for transcriptomics, pretrained on a curated corpus of 545,343 samples spanning human and mouse, including bulk RNA-seq, microarray, and pseudobulked single-cell data, with a focus on **immunology and inflammation (I&I)**. Our main findings are:

- EVA-RNA exhibits power-law scaling up to 300M parameters with no plateau, and pretraining loss improvements consistently translate to downstream performance, contrasting with prior reports of diminishing returns in this domain.

- EVA-RNA learns patient-level representations that achieve strong performance on treatment outcome prediction, patient stratification, and zero-shot target efficacy prediction, tasks where existing foundation models fail to outperform simple baselines.

- EVA-RNA representations integrate gene expression data across technologies and species, in a space spanning relevant biological concepts and presenting high alignment between orthologous genes.

## 2 RESULTS

### 2.1 BENCHMARK PERFORMANCE

We evaluated EVA-RNA on a comprehensive benchmark of I&I tasks spanning drug discovery, preclinical translation, and clinical applications (Table 1). Tasks include prediction of zero-shot target efficacy, gene function, molecular perturbation, cross-species treatment effect, molecular to clinical activity, clinical treatment outcome and the stratification into endotypes. We compare against scGPT (Cui et al., 2024), BulkRNABert (Gélard et al., 2025), and statistical baselines (logistic/ridge regression on expression features and average prediction for gene perturbation).

Table 1: EVA-RNA performance on I&I transcriptomics tasks. Bold and underline indicate the best and 2nd-best models. Values are averaged over five seeds; per-task results with standard deviations are reported in Table S1. Metrics: AUROC (zero-shot, stratification, treatment outcome), AUPRC (gene function), Pearson correlation (perturbation, cross-species, mol. to clinical). Zero-shot target efficacy prediction is not performed with BulkRNABert as the model decoder is not available.

| Task type | 7M | 60M | 300M | scGPT | BulkRNABert | Baseline |
|---|---|---|---|---|---|---|
| Zero-shot target efficacy | 0.655 | 0.619 | **0.693** | 0.539 | – | 0.569 |
| Gene Function | 0.387 | 0.441 | **0.494** | 0.357 | 0.287 | 0.328 |
| Molecular perturbation | 0.511 | 0.544 | **0.547** | 0.454 | 0.475 | 0.171 |
| Cross-Species treatment effect | 0.443 | 0.444 | **0.445** | 0.439 | 0.435 | 0.025 |
| Molecular to clinical activity | 0.383 | 0.429 | **0.434** | 0.360 | 0.312 | 0.427 |
| Stratification into endotypes | 0.767 | **0.799** | 0.786 | 0.706 | 0.695 | 0.762 |
| Clinical treatment outcome | 0.572 | 0.613 | **0.650** | 0.491 | 0.497 | 0.604 |

EVA-RNA achieves state-of-the-art performance on each of the seven task categories. EVA-RNA is especially strong for clinical treatment outcome prediction and stratification into endotypes, where existing foundation models are outperformed by the statistical baseline. The largest improvements over competing foundation models appear in zero-shot target efficacy (0.69 vs. 0.54 for scGPT) and clinical treatment outcome prediction (0.65 vs. 0.49), suggesting that patient-level pretraining on bulk I&I data captures clinically relevant signatures that single-cell models trained on broader data miss. Consistent model improvement from 7M to 300M parameters further validates that our scaling behavior transfers to downstream utility. Benchmark description is provided in Section 3.5; detailed per-task results in Table S1.

These results support our hypothesis that cross-species, multi-technology pretraining produces representations aligned with clinical utility: EVA-RNA succeeds precisely on patient-level tasks where single-species foundation models fail to match simple baselines.

### 2.2 EVA-RNA EXHIBITS CLEAR PRETRAINING SCALING LAWS

To investigate whether scaling laws can emerge in transcriptomic foundation models, we trained EVA-RNA at five model sizes: 7M, 15M, 25M, 60M, and 300M parameters. All models were trained on identical data with consistent hyperparameters, varying only in depth and width, with batch size and learning rate adapted for stability.

Our experiments reveal that EVA-RNA follows predictable power-law scaling (Figure 2a). Fitting $L = aC^{-b}$ to validation loss as a function of compute yields $L = 2.515 \times C^{-0.032}$, indicating that each order of magnitude increase in compute reduces validation loss by approximately 7%. Critically, we observe no sign of plateauing at 300M parameters, suggesting that further model size scaling may yield continued improvements with the current data corpus. This contrasts with results reported in Ho et al. (2024), which observed diminishing returns beyond 100M parameters, and suggests that appropriate data curation (cross-species, multi-technology) and architectural choices may be necessary conditions for observing scaling laws in this domain.

Analysis of intermediate representations reveals a characteristic compression pattern across transformer layers (Figure 2c,d). PCA of sample embeddings shows that layer 29 ($N - 2$) retains multidimensional structure with variance distributed across multiple principal components, while layer 31

($N$) collapses onto the first principal component, concentrating 97.5% of variance. TwoNN intrinsic dimension (Facco et al., 2017) remains high in early layers throughout training but progressively decreases in final layers. This pattern reflects the dual objectives in our training: earlier layers learn rich, distributed gene-gene relationships, while the final layer specializes in gene expression reconstruction. We also see an expansion/compression pattern occurring during training, where the model first expands its representations, then compresses them under the effect of regularization.

Notably, pretraining improvements directly transfer to downstream task performance across evaluation categories (Figure 2b), though scaling profiles vary across tasks. Zero-shot target efficacy, molecular to clinical activity prediction, stratification into endotypes, and clinical treatment outcome show sustained improvement throughout training. Molecular perturbation and cross-species treatment effect show early gains but more modest continued improvement, with higher variance. This pattern suggests that tasks requiring patient-level representations (treatment response, clinical severity) benefit most from extended pretraining, while perturbation prediction tasks may extract most of their value from earlier training.

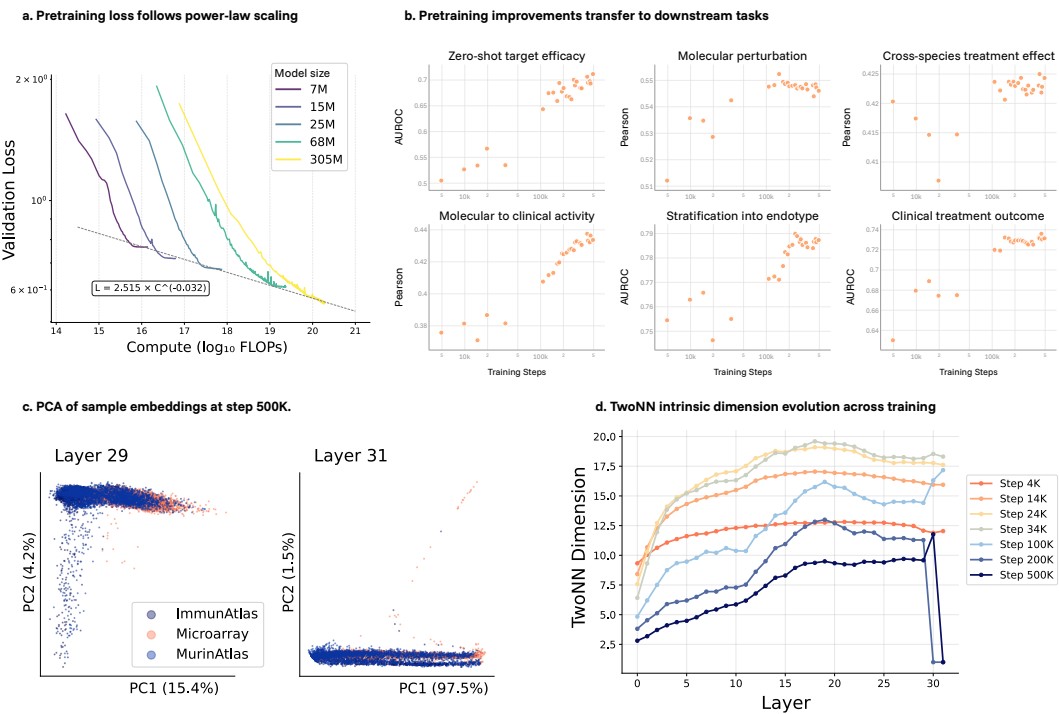

Figure 2: **EVA-RNA exhibits predictable scaling behavior.** (a) Validation loss as a function of compute for five model sizes (7M–300M parameters). Loss follows a power-law relationship with no evidence of plateau. (b) Downstream task performance as a function of training steps across evaluation categories. Tasks improve with continued pretraining. (c) PCA of sample embeddings at layers 29 and 31, i.e. the penultimate (N-2) and final (N) transformer layers, chosen to illustrate the compression pattern. Layer 29 retains multi-dimensional structure, while layer 31 collapses onto the first principal component, reflecting compression toward the reconstruction objective. (d) TwoNN intrinsic dimension across transformer layers at different training checkpoints. Early layers maintain high-dimensional representations, while later layers progressively compress.

## 2.3 CROSS-SPECIES AND CROSS-TECHNOLOGY INTEGRATION

A key challenge in translational research is integrating data across species and technologies. We investigate how EVA-RNA achieves this integration at multiple levels: input gene embeddings, contextualized gene embeddings, and sample embeddings (CLS token) (Figure 3). Understanding this is central: if cross-species training enables scaling and clinical transfer, we should observe the model learning unified representations rather than species- or technology-specific encodings.

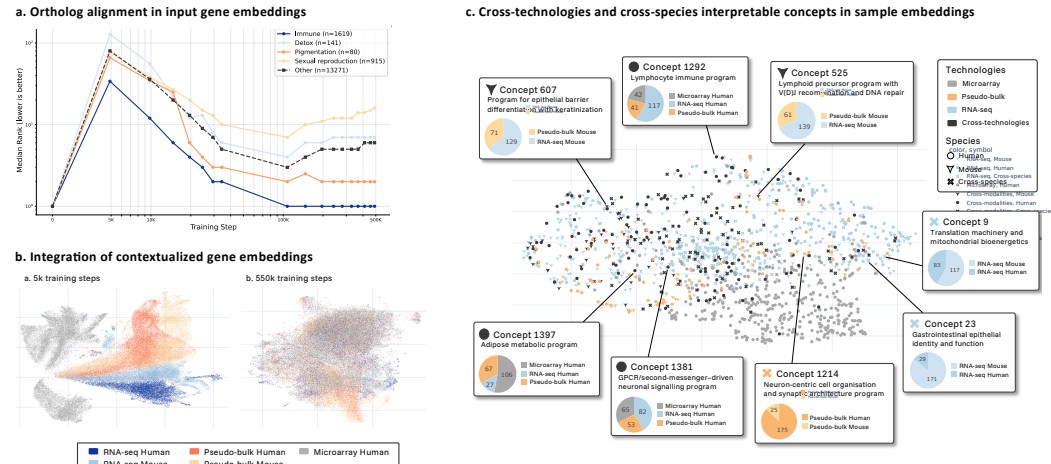

Figure 3: **Cross-technologies and cross-species alignment at multiple levels in EVA-RNA.** (a) Evolution of nearest neighbor median rank between orthologs based on their input embeddings. Immune genes are faster and better aligned than other groups. (b) UMAP of contextualized gene embeddings from layer 30 (N-1) at 5,000 training steps (left) and 550,000 training steps (right), showing progressive integration of human and mouse genes. (c) UMAP of concept vectors extracted from sample embeddings (last CLS token) with TopK SAEs. Colors and markers correspond to the technologies and species among the samples with the highest concept activation. Boxes show examples of concepts with their interpretations and distribution across technologies and species.

**Input gene representations.**   We tracked the evolution of nearest-neighbor median rank between mouse and human orthologs in the input embedding space throughout training (Figure 3a). Starting from an initialization that places orthologs nearby (due to gene embeddings initialization with external knowledge), ranks initially worsen until approximately step 5,000 before steadily improving. This transient degradation likely reflects the model restructuring its latent space during early training, then refining its representations as training goes on. A per-category analysis reveals that immune genes achieve significantly lower final ranks than other gene categories (Bonferroni-corrected Mann-Whitney U tests, $p < 0.05$), with the exception of the Pigmentation group, suggesting that immunity-related genes exhibit particularly strong cross-species alignment.

**Contextualized gene representations.**   Analysis of contextualized gene embeddings from layer 30 ($N − 1$) at early and late training checkpoints reveals a striking pattern (Figure 3b). Early in training, contextualized genes cluster primarily by species and technology. By the final checkpoint, human and mouse genes integrate into a shared embedding space, indicating that the model learns to represent orthologous genes similarly when contextualized by expression patterns.

**Sample representations (CLS token).**   Using TopK SAEs (Gao et al., 2025) and following Claye et al. (2025) to extract interpretable concepts from sample embeddings (Figure 3c), we identified concepts that detect biological signals regardless of technology or species, such as lymphocyte immune programs and epithelial differentiation signatures, suggesting that the model learns technology and species-invariant representations of relevant biological states. Details are provided in Figure S1.

## 3 METHODS

### 3.1 PRETRAINING DATA

EVA-RNA was pretrained on a gene expression atlas curated from public I&I datasets containing 545,343 samples (Table 2). The corpus spans human and mouse, three technologies (bulk RNA-seq, microarray, pseudobulked single-cell), and over 50 tissues and conditions. All datasets underwent quality control and log-normalization: counts were normalized to counts per million (except for

Table 2: Pretraining dataset composition. Weight indicates relative sampling probability during training.

| Dataset | Species | Technology | Samples | Genes | Weight |
|---|---|---|---|---|---|
| ImmunAtlas-seq | Human | Bulk RNA-seq | 42,166 | 39,376 | 0.30 |
| ImmunAtlas-MA | Human | Microarray | 55,564 | 39,376 | 0.20 |
| MurinAtlas | Mouse | Bulk RNA-seq | 437,899 | 26,864 | 0.40 |
| CellxGene human | Human | Pseudobulk | 8,498 | 39,376 | 0.05 |
| CellxGene mouse | Mouse | Pseudobulk | 1,216 | 26,864 | 0.05 |
| **Total** | | | **545,343** | **66,240** | **1.00** |

microarray), then log-transformed via $\log_2(x+1)$. During training, datasets were sampled with weights emphasizing bulk RNA-seq while maintaining cross-species and single-cell representation.

EVA-RNA uses a multi-species gene vocabulary of 66,240 human and mouse NCBI Gene IDs, chosen over gene symbols to avoid ambiguity from synonyms and naming inconsistencies. The vocabulary was filtered to genes present in bulk RNA-seq datasets.

## 3.2 MODEL ARCHITECTURE

EVA-RNA is a 300M parameters model consisting of 32 transformer layers (indexed from 0 to 31 throughout the paper), 768 hidden dimensions, 12 attention heads, and 3072-dimensional feed-forward layers. We employ pre-layer normalization with residual scaling by $1/\sqrt{2L}$ where $L$ is the number of layers (Radford et al., 2019). Gene expression values are embedded through a multi-layer perceptron followed by layer normalization, whose output is added to the gene token embedding. Each input sequence is prepended with a CLS token whose final representation serves as the sample-level embedding.

**Gene embedding initialization with external knowledge.** Inspired by other transcriptomics foundation model approaches (Yang et al., 2024; Kalfon et al., 2025), EVA-RNA leverages external knowledge via precomputed gene embeddings coming from five sources (used for initialization only, then fine-tuned during pretraining): (1) scGPT gene embeddings capturing single-cell co-expression patterns (Cui et al., 2024); (2) ESM-2 (650M) protein embeddings encoding amino acid sequence information (Lin et al., 2023); (3) text embeddings from NCBI gene descriptions (Sayers et al., 2025); (4) text embeddings from UniProt protein descriptions (Bateman et al., 2025); and (5) knowledge graph embeddings (KGE) computed with RotatE method (Sun et al. (2019)). All external sources matrices are reduced via per-source PCA, concatenated, and passed through a two-layer MLP to produce the final embedding (Figure 4a-b). A fallback matrix (initialized at random) is added to account for genes that are not supported by any external sources. Ablation studies confirm that combining all five sources yields lower validation loss than any individual source (Figure 4d), and this translates to downstream task performance across all evaluation categories (Figure 4e).

## 3.3 TRAINING OBJECTIVES

**ZINB decoder for masked gene expression.** Gene expression data exhibit high sparsity and overdispersion relative to the Poisson distribution. To capture these properties, EVA-RNA employs a Zero-Inflated Negative Binomial (ZINB) decoder (Lopez et al., 2018) that models the conditional distribution of gene expression. For each masked position, the decoder predicts three parameters from the contextualized embedding: mean $\mu$, dispersion $\theta$, and zero-inflation probability $\pi$. The training objective minimizes the negative log-likelihood over masked positions

$$\mathcal{L}_{\text{ZINB}} = -\frac{1}{|\mathcal{M}|} \sum_{i \in \mathcal{M}} \log P(x_i \mid \mu_i, \theta_i, \pi_i) \tag{1}$$

where $\mathcal{M}$ denotes the set of masked gene indices.

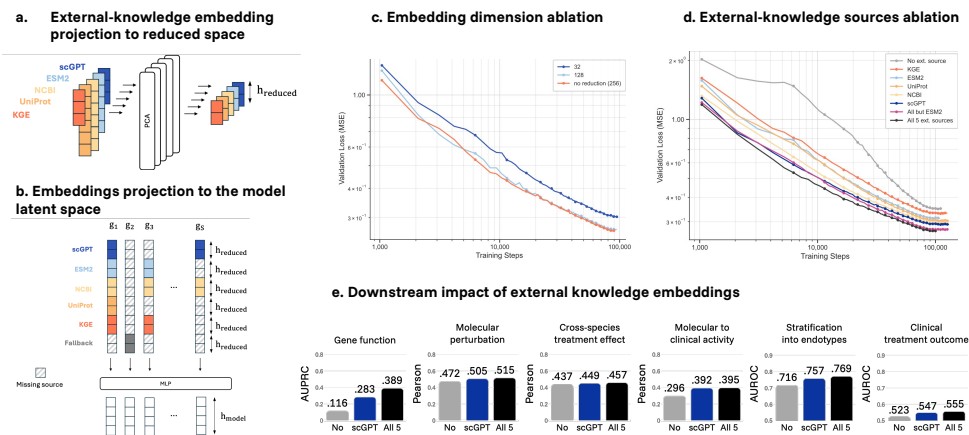

Figure 4: **External knowledge integration for gene embeddings.** (Details in Section 3) (a) At initialization, all external knowledge embeddings are reduced to a shared dimension via a per-source PCA. (b) For a given gene $g_i$, reduced embeddings from all sources are concatenated (channels are zeroed out if $g_i$ is not supported by the source, e.g., $g_3$ is not supported by scGPT and UniProt) and passed through an MLP to generate the final embedding. On the figure, $g_2$ is not supported by any external knowledge source, hence its embedding is only derived from the fallback matrix. (c) Using a reduced vocabulary size provides similar training dynamics as full-size while saving training parameters. (d) Initializing with all 5 external knowledge sources yields the best performance both in convergence and final validation loss. (e) External knowledge advantage translates from pretraining to downstream tasks, with clear increase of performance across all task categories.

**CLS token reconstruction.** The CLS token embedding is trained through an auxiliary objective that enforces global sample-state awareness. Given the contextualized CLS embedding and input (non-contextualized) gene embeddings, the model learns to reconstruct all gene expressions via a two-layer MLP decoder with MSE loss. This objective encourages the CLS token to learn a compressed representation of the sample state. The total loss combines both objectives

$$\mathcal{L}_{\text{total}} = (1 - \lambda)\mathcal{L}_{\text{ZINB}} + \lambda\mathcal{L}_{\text{CLS}}, \quad \lambda = 0.5. \tag{2}$$

## 3.4 PRETRAINING

We employ a curriculum learning schedule where the masking ratio decreases linearly from 95% to 15% over 500,000 training steps, as suggested in Wettig et al. (2023) and Ankner et al. (2024). We hypothesize that high masking ratios early in training force the model to learn robust coarse-grained representations before transitioning to scenarios with richer input context.

The maximum sequence length increases linearly from 600 to 1,800 genes over 400,000 steps following a 1,000-step warmup period, improving training efficiency while enabling the model to handle variable-length inputs.

We apply mixup with probability 1.0 and $\alpha = 1.0$, linearly interpolating expression profiles between randomly paired samples within each batch. This aggressive mixup setting helps the model generalize across technologies and species.

For pretraining, we use AdamW (Loshchilov & Hutter, 2017) with $\beta_1 = 0.9$, $\beta_2 = 0.999$, and weight decay 0.05. The learning rate follows a warmup-cosine schedule: linear warmup to $1.7 \times 10^{-4}$ over 1,000 steps, then cosine decay to $5 \times 10^{-6}$ over 250,000 steps. Training uses mixed-precision (bfloat16) with gradient clipping to 1.0. The 300M model was trained for approximately 4,000 GPU-hours on A100s.

## 3.5 Evaluation Benchmark

We curated a benchmark of 35 tasks spanning the drug development pipeline for I&I, organized into 3 categories. Tasks were selected to cover the pipeline end-to-end, from target identification through preclinical translation to clinical decision-making, prioritizing settings where foundation model representations can be directly evaluated.

**Discovery.** Zero-shot target efficacy prediction evaluates whether perturbing a drug's molecular target shifts patient transcriptomes toward healthy states. We leverage decoder gradients to perform *in silico* gene perturbations without any perturbation-labeled training data, following Bjerregaard et al. (2025). Gene function prediction assesses whether gene embeddings capture functional relationships through multi-label classification of disease associations, Gene Ontology terms, cell type markers, and pathway membership.

**Preclinical.** Molecular perturbation prediction evaluates the model's ability to predict full transcriptomic changes following therapeutic intervention. Cross-species treatment effect tests whether perturbation patterns learned from mouse models transfer to human disease, with models fine-tuned on mouse data and evaluated on human samples.

**Clinical.** Molecular to clinical activity predicts clinical severity indices and disease activity score (e.g. EASI, PASI, Mayo score) from gene expression across multiple I&I diseases. Stratification into endotypes classifies rheumatoid arthritis pathotypes from blood or synovial tissue. Clinical treatment outcome prediction evaluates binary prediction of therapeutic response across multiple drugs.

All tasks use subject-level data splitting to prevent leakage, evaluation across five random seeds, and comparison against statistical baselines (ridge/logistic regression/average perturbator) and foundation models (scGPT, BulkRNABert). Expression data underwent $\log_2(\text{CPM}+1)$ normalization with K-best feature selection (k=8000) for linear probes.

## 4 Discussion

We introduced EVA-RNA, a cross-species transcriptomic foundation model that demonstrates clear scaling laws and learns integrated representations across species and technologies. Our results challenge the notion that gene expression models cannot benefit from scale: with appropriate data curation and training methodology, EVA-RNA exhibits predictable power-law scaling up to 300M parameters with no sign of plateauing, contrasting with prior reports of diminishing returns in this domain (Ho et al., 2024).

Several design choices appear critical to these results. First, cross-species training on both human and mouse data may provide richer supervision than single-species corpora, as the model must learn conserved biological programs to succeed on both. Second, multi-technology integration forces the model to learn representations invariant to platform-specific biases, potentially improving generalization. Third, domain focus on I&I where shared pathogenic mechanisms create natural transfer learning opportunities may be more effective than training general-purpose models on heterogeneous biological contexts.

Our analysis reveals that pretraining improvements transfer to downstream task performance, though scaling profiles vary across tasks. Zero-shot target efficacy, molecular to clinical activity, stratifica-

tion into endotypes, and clinical treatment outcome show sustained improvement throughout training, validating alignment between our pretraining objective and clinically relevant representations. Molecular perturbation and cross-species treatment effect show early gains but more modest continued improvement, with higher variance. This pattern suggests that tasks requiring patient-level representations benefit most from extended pretraining, while perturbation prediction tasks may extract most of their value from earlier training.

**Limitations.** EVA-RNA operates on bulk and pseudobulk representations, potentially obscuring cell-type-specific responses critical for understanding therapeutic mechanisms. The model's domain focus on immunology may limit generalization to other therapeutic areas. Finally, while we demonstrate strong benchmark performance, prospective validation in clinical settings remains necessary to establish translational utility.

**Conclusion.** EVA-RNA demonstrates that transcriptomic foundation models can exhibit predictable scaling behavior when trained on cross-species, multi-technology data with appropriate methodology. By releasing an open version of EVA-RNA (`https://huggingface.co/ScientaLab/eva-rna`), we aim to accelerate translational research in immune-mediated diseases and provide with a foundation to develop more effective therapies for conditions affecting hundreds of millions worldwide.

### MEANINGFULNESS STATEMENT

Meaningful representations of life should capture conserved biological programs that transcend experimental artifacts and enable translation from model organisms to human disease. EVA-RNA learns such representations by jointly modeling human and mouse transcriptomes across technologies, demonstrating that orthologs progressively align in embedding space and that the model encodes shared biological concepts. These representations directly enable drug development tasks: predicting treatment response, stratifying patients, and translating preclinical findings, bridging molecular measurements and clinical utility. By establishing scaling laws for transcriptomic foundation models, we provide evidence that further scaling may yield increasingly meaningful representations of immunology.

### ACKNOWLEDGMENTS

This project was partially supported by computational and storage resources from the GENCI at IDRIS, thanks to the grant 2025-AD010316294 on the supercomputer Jean Zay's A100 and H100 partitions.

### LLM USAGE POLICY

We used Claude (by Anthropic) as a coding assistant to help write and debug experimental code and data processing scripts. For manuscript preparation, we used Claude and Grammarly to improve grammar, clarity, and writing quality. All research ideas, experimental design, scientific claims, and results were developed independently by the authors. All LLM-generated code and text were reviewed, validated, and verified by the authors.

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

# SUPPLEMENTARY MATERIAL

## DETAILED BENCHMARK RESULTS

Table S1: EVA-RNA performance on all I&I benchmark tasks (mean $\pm$ std). Bold/underline: best/second-best. Arrows: transfer direction. BulkRNABert cannot perform zero-shot prediction as the decoder is unavailable.

| Task | 7M | 60M | 300M | scGPT | BulkRNABert | Baseline |
|---|---|---|---|---|---|---|
| **Zero-shot target efficacy (AUROC)** | | | | | | |
| Target efficacy | 0.66 | 0.62 | **0.69** | 0.54 | – | 0.57 |
| **Gene function (AUPRC)** | | | | | | |
| CellType | $0.37 \pm 0.23$ | $0.45 \pm 0.22$ | $0.41 \pm 0.19$ | $\mathbf{0.48 \pm 0.24}$ | $0.34 \pm 0.21$ | $0.39 \pm 0.23$ |
| Disease | $0.33 \pm 0.31$ | $0.31 \pm 0.36$ | $\mathbf{0.46 \pm 0.34}$ | $0.24 \pm 0.32$ | $0.22 \pm 0.26$ | $0.25 \pm 0.30$ |
| GO | $0.40 \pm 0.29$ | $0.43 \pm 0.30$ | $\mathbf{0.47 \pm 0.27}$ | $0.34 \pm 0.28$ | $0.30 \pm 0.22$ | $0.29 \pm 0.28$ |
| Reactome | $0.50 \pm 0.29$ | $0.58 \pm 0.27$ | $\mathbf{0.64 \pm 0.29}$ | $0.36 \pm 0.22$ | $0.30 \pm 0.23$ | $0.32 \pm 0.25$ |
| WikiPathways | $0.33 \pm 0.22$ | $0.42 \pm 0.24$ | $\mathbf{0.49 \pm 0.27}$ | $0.37 \pm 0.21$ | $0.27 \pm 0.20$ | $0.38 \pm 0.23$ |
| **Molecular perturbation (Pearson)** | | | | | | |
| Anti-TNF (IBD mice) | $0.54 \pm 0.13$ | $0.66 \pm 0.17$ | $\mathbf{0.70 \pm 0.16}$ | $0.37 \pm 0.19$ | $0.49 \pm 0.18$ | $0.59 \pm 0.21$ |
| Adalimumab (HS→Pso) | $\mathbf{0.48 \pm 0.01}$ | $0.46 \pm 0.01$ | $0.45 \pm 0.01$ | $0.41 \pm 0.02$ | $0.41 \pm 0.03$ | $-0.19 \pm 0.02$ |
| Adalimumab (Pso→HS) | $0.27 \pm 0.04$ | $0.29 \pm 0.01$ | $\mathbf{0.32 \pm 0.00}$ | $0.23 \pm 0.01$ | $0.24 \pm 0.02$ | $-0.13 \pm 0.00$ |
| Rituximab (SjD blood) | $0.77 \pm 0.02$ | $\mathbf{0.78 \pm 0.03}$ | $0.77 \pm 0.02$ | $0.75 \pm 0.04$ | $0.74 \pm 0.02$ | $0.60 \pm 0.04$ |
| Rituximab (SjD salivary) | $0.50 \pm 0.11$ | $\mathbf{0.52 \pm 0.10}$ | $0.50 \pm 0.10$ | $0.51 \pm 0.10$ | $0.49 \pm 0.12$ | $-0.01 \pm 0.09$ |
| **Cross-species treatment effect (Pearson)** | | | | | | |
| Dupilumab | $0.47 \pm 0.00$ | $0.48 \pm 0.01$ | $0.48 \pm 0.00$ | $0.47 \pm 0.00$ | $\mathbf{0.48 \pm 0.00}$ | $0.05 \pm 0.00$ |
| TNFi RA | $0.41 \pm 0.00$ | $\mathbf{0.41 \pm 0.00}$ | $0.41 \pm 0.00$ | $0.41 \pm 0.00$ | $0.39 \pm 0.00$ | $0.00 \pm 0.00$ |
| **Molecular to clinical activity (Pearson)** | | | | | | |
| Blood IgA | $0.37 \pm 0.11$ | $\mathbf{0.39 \pm 0.09}$ | $0.37 \pm 0.11$ | $0.28 \pm 0.13$ | $0.18 \pm 0.12$ | $0.39 \pm 0.10$ |
| Blood IgG | $0.47 \pm 0.07$ | $0.51 \pm 0.07$ | $0.51 \pm 0.10$ | $0.41 \pm 0.06$ | $0.25 \pm 0.15$ | $\mathbf{0.54 \pm 0.09}$ |
| Digestive GHAS7 | $0.60 \pm 0.05$ | $0.61 \pm 0.04$ | $0.57 \pm 0.02$ | $0.58 \pm 0.05$ | $0.55 \pm 0.07$ | $\mathbf{0.61 \pm 0.03}$ |
| Digestive SES-CD | $0.39 \pm 0.07$ | $0.45 \pm 0.10$ | $\mathbf{0.46 \pm 0.09}$ | $0.39 \pm 0.07$ | $0.35 \pm 0.07$ | $0.44 \pm 0.08$ |
| ESSDAI Bio | $0.35 \pm 0.10$ | $0.40 \pm 0.08$ | $\mathbf{0.41 \pm 0.16}$ | $0.33 \pm 0.08$ | $0.13 \pm 0.15$ | $0.40 \pm 0.08$ |
| Endoscopic Mayo | $0.64 \pm 0.13$ | $\mathbf{0.67 \pm 0.13}$ | $0.64 \pm 0.15$ | $0.65 \pm 0.15$ | $0.60 \pm 0.15$ | $0.66 \pm 0.12$ |
| HBI | $0.14 \pm 0.05$ | $0.15 \pm 0.08$ | $\mathbf{0.24 \pm 0.07}$ | $0.14 \pm 0.06$ | $0.12 \pm 0.08$ | $0.19 \pm 0.06$ |
| Nancy Index | $0.76 \pm 0.11$ | $0.75 \pm 0.07$ | $0.73 \pm 0.07$ | $0.75 \pm 0.09$ | $0.71 \pm 0.14$ | $\mathbf{0.76 \pm 0.12}$ |
| RA SJC28 | $0.27 \pm 0.22$ | $0.36 \pm 0.21$ | $\mathbf{0.44 \pm 0.16}$ | $0.29 \pm 0.20$ | $0.24 \pm 0.18$ | $0.36 \pm 0.20$ |
| RA TJC28 | $0.23 \pm 0.24$ | $0.21 \pm 0.21$ | $0.27 \pm 0.17$ | $0.24 \pm 0.20$ | $0.19 \pm 0.21$ | $\mathbf{0.28 \pm 0.25}$ |
| SCCAI | $0.52 \pm 0.04$ | $\mathbf{0.61 \pm 0.05}$ | $0.59 \pm 0.07$ | $0.50 \pm 0.06$ | $0.49 \pm 0.09$ | $0.57 \pm 0.05$ |
| Skin EASI | $\mathbf{0.32 \pm 0.19}$ | $0.28 \pm 0.16$ | $0.29 \pm 0.20$ | $0.29 \pm 0.16$ | $0.24 \pm 0.15$ | $0.30 \pm 0.22$ |
| Skin PASI | $0.18 \pm 0.15$ | $0.17 \pm 0.22$ | $\mathbf{0.26 \pm 0.24}$ | $0.10 \pm 0.16$ | $0.21 \pm 0.08$ | $0.25 \pm 0.22$ |
| Skin SCORAD | $0.11 \pm 0.16$ | $0.21 \pm 0.11$ | $\mathbf{0.27 \pm 0.14}$ | $0.07 \pm 0.13$ | $0.09 \pm 0.11$ | $0.21 \pm 0.19$ |
| **Stratification into endotypes (AUROC)** | | | | | | |
| RA blood | $0.64 \pm 0.06$ | $\mathbf{0.66 \pm 0.12}$ | $0.65 \pm 0.12$ | $0.52 \pm 0.07$ | $0.58 \pm 0.08$ | $0.63 \pm 0.12$ |
| RA joint | $0.90 \pm 0.02$ | $0.91 \pm 0.02$ | $\mathbf{0.92 \pm 0.01}$ | $0.89 \pm 0.02$ | $0.81 \pm 0.03$ | $0.90 \pm 0.01$ |
| **Clinical treatment outcome (AUROC)** | | | | | | |
| IBD, adalimumab | $0.46 \pm 0.07$ | $0.56 \pm 0.13$ | $\mathbf{0.76 \pm 0.07}$ | $0.40 \pm 0.12$ | $0.37 \pm 0.16$ | $0.48 \pm 0.17$ |
| IBD, infliximab | $0.53 \pm 0.19$ | $0.55 \pm 0.29$ | $\mathbf{0.80 \pm 0.24}$ | $0.50 \pm 0.13$ | $0.48 \pm 0.18$ | $0.62 \pm 0.10$ |
| IBD, ADA→IFX | $0.65 \pm 0.08$ | $0.56 \pm 0.08$ | $0.50 \pm 0.05$ | $0.59 \pm 0.05$ | $0.64 \pm 0.07$ | $\mathbf{0.66 \pm 0.04}$ |
| IBD, IFX→ADA | $\mathbf{0.63 \pm 0.02}$ | $0.57 \pm 0.06$ | $0.57 \pm 0.04$ | $0.61 \pm 0.04$ | $0.57 \pm 0.02$ | $0.62 \pm 0.02$ |
| IBD, vedo, seq→arr | $0.52 \pm 0.11$ | $0.62 \pm 0.03$ | $\mathbf{0.65 \pm 0.04}$ | $0.47 \pm 0.09$ | $0.47 \pm 0.07$ | $0.61 \pm 0.01$ |
| IBD, vedo, arr→seq | $\mathbf{0.64 \pm 0.02}$ | $0.60 \pm 0.05$ | $0.63 \pm 0.05$ | $0.37 \pm 0.03$ | $0.44 \pm 0.07$ | $0.64 \pm 0.02$ |

LATENT SPACE INTERPRETABILITY

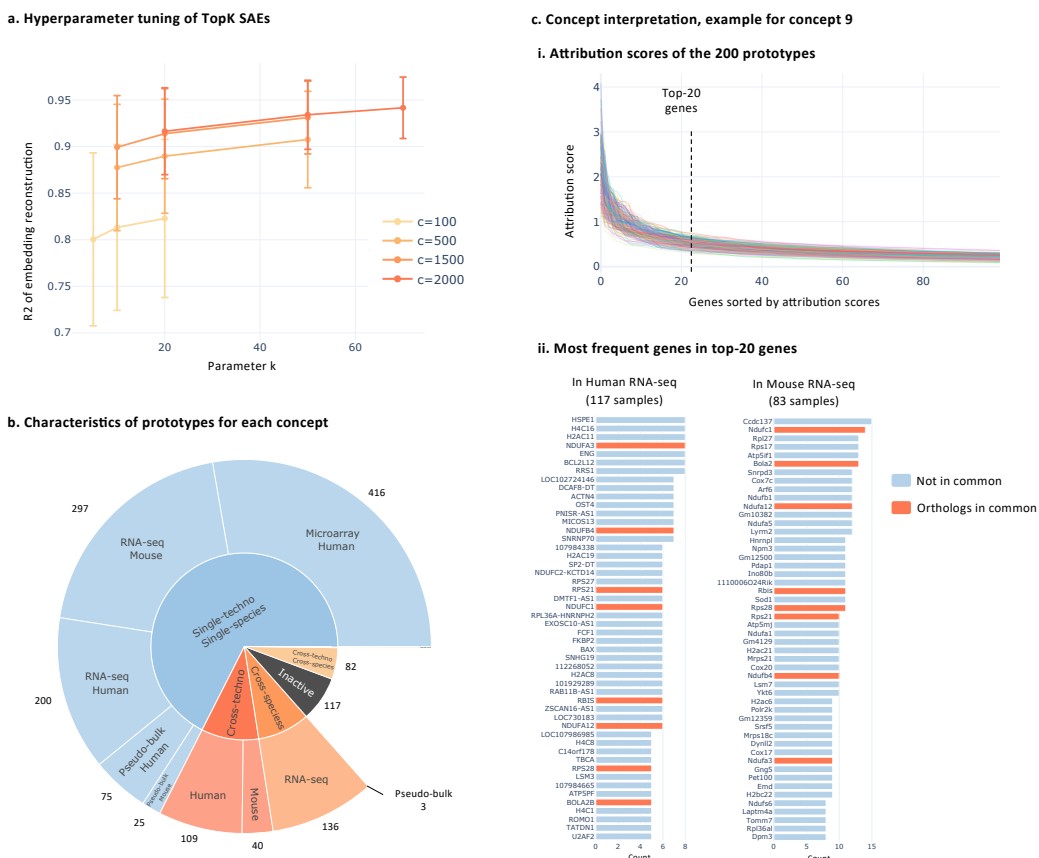

Figure S1: Latent space interpretability with TopK Sparse Auto-encoders Gao et al. (2025). a. Hyperparameter search for topK-SAEs for concept extraction in contextualized sample representations (last CLS token). Embedding reconstruction error at different $c$ and $k$. A score of 1 means that the embeddings are reconstructed perfectly from the concept activations, while a score of 0 means that the reconstruction is no better than the mean. b. Distribution of concepts in terms of species and technologies represented in the 200 samples with the highest concept activation ("prototypes"). c. Example of attribution results for concept 9. (a) Attribution scores (using Integrated Gradient Sundararajan et al. (2017)) from the highest score to the lowest for each prototype. For all prototypes, the top 20 genes have a high attribution score. (b) Most frequent genes in the top-20 genes of each prototype, by species. Given the mapping of orthologous genes between human and mouse, several genes that are most frequently important in human samples are also important in mouse samples, suggesting a shared interpretation across the species.

