# OpenReview forum: "EVA-RNA: A Scaling Cross-Species Transcriptomic Foundation Models for Immunology & Inflammation"
_ICLR.cc/2026/Workshop/LMRL — ICLR 2026 Workshop LMRL Poster_

### Official Review · Reviewer_eKYA · 2026-02-23
**Interesting analysis of multi-species foundation model pretraining and scaling**

**Rating:** 6
**Confidence:** 4

**Review:**

the proposed methods, EVA-RNA learns representations by jointly modeling human and mouse transcriptomes, demonstrating that orthologs progressively align in embedding space and that the model encodes shared biological concepts. The authors suggest that such representations help in drug development tasks like predicting treatment response, stratifying patients, and translating preclinical findings, bridging molecular measurements and clinical utility. They derive scaling laws for these foundation models. The paper needs to provide much more detail to effectively consider such claims, particularly in the light of the drug discovery applications possible. That would make this a very nice paper eventually.

---

### Official Review · Reviewer_4pFJ · 2026-02-24

**Rating:** 8
**Confidence:** 2

**Review:**

The authors present EVA-RNA, a transcriptomic fondation model, trained on human and mouse data which is focused on immunology and inflammation. Results indicate power-law-like scaling with model parameters and the ability for a foundation model to match or exceed baselines on downstream tasks at which previous foundation models struggled.

Pros:
- The paper is well written and adequately concise.
- The work makes a strong case of progressing in the domain of immunology and inflammation and it is appreciated the authors clearly define this scope (starting with the title) and do not position the model as entirely general.
- The integration of prior knowledge is a great use of the flexibility deep learning and transformers offer, including ablations.

Cons:
- A critical lack of variability estimates (and/or error bars) for results in the main text. Progress on some tasks is very small and likely not statistically significant; including some variability estimates would already go a long way in conveying to the reader on which tasks solid progress is made and where there is not.
- There are aspects which differ from previous foundation models, but which are not ablated. For example, a common issue in this line of research concerns the pretraining data mixture - if this is not controlled for, performance comparisons between models are confounded and it's not entirely clear how modeling deicsions affect the final model ranking. Moreover, besides a general versus an I&I-focused data corpus, theres also single versus multi-species, dual pretraining objective, masking schedule, etc.
- I found myself skipping ahead to the methods before returning to the results; I think it would have been better if the authors had stuck  to the usual paper structure.

Overall, I believe the paper forms a strong contribution to the workshop.

---

### Official Review · Reviewer_jGPT · 2026-02-24
**Review: EVA-RNA**

**Rating:** 8
**Confidence:** 3

**Review:**

1. **Summary**
The authors train a transcriptomic foundation model with a focus on immunology and inflammation. The model is notably trained on data from humans and mice, utilizing bulk RNA-seq, pseudobulk, and microarray data . The model appears to follow power-law scaling up to 300M parameters, pushing beyond previous reports of diminishing returns for single-species models.

2. **Interpretability and Scaling**
* The interpretability results on the sample representations appear to show that the gene embeddings become integrated across modalities and species.
* The authors show that the model's downstream performance increases with increased training, and that the intrinsic dimensionality of the higher layers is reduced (as estimated using variance explained from the first 2 principal components and the TwoNN Dimension)

3. **Critiques & Weaknesses**
* **Benchmark Selection Rationale**: The model appears to perform well in benchmark evaluation; however, the selection criterion for these benchmarks is a bit unclear. It would be interesting to know if there are other metrics that could be considered and why they were excluded.
* **Baseline Fairness**: Several of these metrics explicitly test cross-species transferability, which disadvantages the single-species baseline models.
* **Evaluation Methods**: Using experimental gene perturbations instead of in-silico for evaluation would be more useful.
* **Figure 2 Feedback**: It is unclear why the authors chose Layer 29 and 31 for Fig 2c. In my opinion, it would be more informative just to plot the variance explained for the first 2 PCs against the layer index, as is done in Fig 2d. Overall, this analysis is interesting, but it is not clear what this adds to the paper (although the representation geometry of FMs is an interesting avenue for research).

**Limitations (as noted by authors)**: The model operates on bulk and pseudobulk representations, which might obscure important cell-type-specific responses. Additionally, its specialized focus on immunology may limit its generalization to other therapeutic domains, and prospective validation in real clinical settings is still required.

4. **Questions for the Authors**
* From a biological perspective, is there a reason why immunity-related genes exhibit particularly strong cross-species alignment? Answering questions like this would help to contextualize the results.
* Minor Nits: What is "AIDO" on line 78?

---

### Meta-Review · Area_Chair_UjX1 · 2026-02-28

**Recommendation:** Accept (Poster)
**Confidence:** 4

**Metareview:**

Accept

---

### Decision · Program_Chairs · 2026-03-02

**Decision:**

Accept (Poster)

**Comment:**

Please see the meta-review.